# Arctic Sea Ice Lead Detection from Chinese HY-2B Radar Altimeter Data

**Wenqing Zhong** [1,2]**, Maofei Jiang** [1,*] **, Ke Xu** [1] **and Yongjun Jia** [3,4]

1   The CAS Key Laboratory of Microwave Remote Sensing, National Space Science Center, Chinese Academy of Sciences (CAS), Beijing 100190, China
2   University of Chinese Academy of Sciences, Beijing 100049, China
3   Key Laboratory of Space Ocean Remote Sensing and Application, MNR, Beijing 100049, China
4   National Satellite Ocean Application Service, Beijing 100082, China
*   Correspondence: jiangmaofei@mirslab.cn

**Abstract:** Sea ice thickness is one of the essential characteristics of sea ice. Sea ice lead detection is the key to sea ice thickness estimation from radar altimetry data. This research studies ten different surface type classification methods, including supervised learning, unsupervised learning, and threshold methods, being applied to the HY-2B radar altimeter data collected in October 2019 in the Arctic Ocean. The Sentinel-1 Synthetic Aperture Radar (SAR) images were used for training and validation of the classifiers. Compared with other classifiers, the supervised Bagging ensemble learning classifier showed excellent and robust performance with overall accuracy up to 95.69%. In order to assess the performance of the Bagging classifier in practical applications, lead fractions from January 2019 to March 2021 based on the HY-2B radar altimeter data were mapped using the trained Bagging classifier and compared to the CryoSat-2 L2I data product. The results of the lead fraction showed the monthly variability of ice lead, and the ice lead had a reasonable spatial distribution and was consistent with CryoSat-2 L2I data products. According to these results, the Bagging classifier can provide an essential reference for future studies of Arctic sea ice thickness and sea level estimation from HY-2B radar altimeter data.

**Keywords:** HY-2B; radar altimeter; sea ice lead; classification method; sea ice thickness

## 1. Introduction

As one of the essential characteristics of sea ice, sea ice thickness is particularly significant and sensitive to air/ice/sea interactions and directly determines the processes and speeds of substance and energy exchange between the ocean and atmosphere [1,2]. In addition, it dominates the thermodynamic and dynamic properties of sea ice [3]. For example, it affects the drift, deformation, freezing, and melting process of sea ice and then feeds back to the global climate system, which causes a series of changes in climate and environmental parameters related to human survival [4]. Scientific evidence shows that, due to the impact of global climate change, sea ice thickness is steadily decreasing in the Arctic [5,6]. Consequently, research on monitoring sea ice thickness is significant in responding to global climate change.

Sea ice thickness is one of the geophysical parameters most challenging to measure. Over the past few decades, satellite radar and laser altimeter data in the polar region have been widely used to estimate sea ice thickness and volume [7–11]. Compared with laser altimeters, radar altimeters have the advantage of being unaffected by clouds and fog due to microwave signals. The principle of using an altimeter to calculate sea ice thickness is to obtain the sea ice freeboard and then obtain the sea ice thickness using Archimedes' law [12], where the sea ice freeboard is the difference between sea ice height and sea surface height (SSH). However, estimating SSH in the Arctic is challenging since much of the ocean is covered with sea ice. Satellite altimeter measurements use sea ice

openings called ice leads, and consider them as instantaneous SSH [13]. They then obtain the SSH corresponding to sea ice by interpolating the height of the leads. Thus, accurate lead detection of radar altimeters is an essential prerequisite for reversing sea ice thickness. Incorrect classification may increase the uncertainty of sea ice freeboard and thickness.

Several classification methods have been developed to detect ice leads, ocean (open water), and sea ice. These waveform classification methods can be primarily divided into two categories: traditional threshold method and machine learning classification method [13]. Most methods are based on waveform features, which describe the unique features of different targets. The threshold method is based on experience to set the threshold of waveform features to distinguish waveform types. Historically, this method has been widely used in studying sea ice thickness [14–16]. For example, Zygmuntowska et al. used waveform maximum, PP, Leading Edge Width (LEW), and Trailing Edge width (TEW) to study waveform classification of Cryostat-2 radar altimeter over Arctic sea ice [17]. Recently, an improved threshold method based on iteration has been gradually applied to altimeter waveform classification [18–20]. However, the threshold method could lead to over-estimate leads, which can increase the bias of sea ice thickness [13,21].

Recently, machine learning algorithms have been gradually applied in the classification of altimeter waveforms, which can improve the accuracy of the classification compared to the threshold method [13,19,22]. However, supervised machine learning methods require accurate ground truth values for altimeter waveforms. Recently, satellite optical and SAR images, which have higher resolution, have been used to provide ground truth values for radar altimeter waveforms. Lee et al. used MODIS images obtained by Terra and Aqua satellites to provide labels for sea ice and leads for the CryoSat-2 radar altimeter and then used machine learning methods for classification [21]. In addition, Müller et al. used SAR images to validate the performance of unsupervised machine learning in Envisat and SARAL altimeter lead detection [18]. However, these studies have not considered the off-nadir leads, as they only considered the pixel type corresponding to the track coordinates as ground truth. In fact, the contribution from the ice can be overshadowed by the off-nadir lead [23]. Therefore, there is no concrete understanding of waveform returns of off-nadir leads.

In addition, most of the research on lead detection and sea ice thickness inversion is aimed at the CryoSat-2 altimeter. The Cryosat-2 altimeter is a synthetic aperture radar altimeter whose azimuthal high resolution gives it a distinct advantage for sea ice thickness inversion. The HY-2B altimeter, like Envisat-2, is a conventional pulse-limited footprint radar altimeter. Although there are some shortcomings compared with Cryosat-2, it can still provide measurement data for studying sea ice thickness and polar sea level changes. However, there are few studies on the detection of ice leads and inversion of sea ice thickness for traditional altimeters, especially the HY-2B altimeter. Dong et al. and Zhang et al. used the L2 products of the HY-2B altimeter to detect leads from sea ice [24,25]. However, the L2 products of the HY-2B radar altimeter only include the data from the sub-optimal maximum likelihood estimate (SMLE) package, which lacks the data from the off center of gravity (OCOG) package [26]. In fact, the OCOG mode is selected when the mode compatible tracker (MCT) of the HY-2B radar altimeter detects a larger amplitude difference between the power spectrum waveform and the track window [26]. Instead, SMLE mode will be selected. Due to the calm water surface, a sea ice lead can be approximated as a specular reflection, which makes the echo power have a large amplitude. Therefore, the lead data of the HY-2B radar altimeter are mainly contained in the OCOG package. The robustness and reliability of the study may be reduced when only using the data from the SMLE tracker. Thus, to use the HY-2B radar altimeter to compute sea ice freeboard and sea ice thickness accurately, it is necessary to study the waveform classification of the HY-2B radar altimeter (especially the detection of ice leads).

In this paper, we used the L1B product of the HY-2B altimeter in October 2019, which included both the OCOG and SMLE package data, to study waveform classification methods. This study compared the classification performance of improved threshold, unsupervised

machine learning, and supervised learning methods. In addition, this study first investigated the statistical waveforms of different sea ice concentrations within the footprint of the HY-2B radar altimeter based on Sentinel-1 SAR images. Furthermore, this study used the best classifier to map ice lead fractions from January 2019 to March 2021 and compared these results with CryoSat-2 L2I level data products.

This paper is structured as follows. In Section 2, the data used in this study and the processing method are introduced. Section 3 presents and comments on the results obtained with the presented method and compares them with the products from other missions. Finally, the conclusions are summarized in Section 4.

## 2. Materials and Methods

### 2.1. Data

#### 2.1.1. HY-2B

HY-2B, launched on 26 October 2018, is China's ocean dynamic environment satellite. A dual-frequency traditional radar altimeter (Ku and C bands) is onboard the HY-2B satellite [27]. The ground pulse-limited footprint diameter of the C band is approximately 10 km, and the Ku band is 1.9 km. Moreover, the Ku band is often used in practical ranging, and the primary purpose of the C band is to provide dual-frequency ionospheric correction for Ku band ranging. Therefore, in this study, we only used the Ku band data of HY-2B. The altimeter uses the conventional pulse-limited method to measure sea surface height, significant wave height, and sea surface wind speed. Due to the fact that the data coverage can reach 80.6°N–80.6°S, which can cover most of the sea ice in polar regions, the HY-2B can be used for sea ice measurements in polar regions.

The HY-2B radar altimeter has two tracking models cooperating in parallel: sub-optimal maximum likelihood estimate (SMLE) and off center of gravity (OCOG). The tracker of SMLE is model-dependent. Therefore, the SMLE tracker often loses track in areas where topography changes dramatically [26,27]. To solve this problem, the HY-2B radar altimeter uses a compatible tracker, which includes SMLE and OCOG tracker units. When the amplitude difference between the power spectrum waveform and the track window is large, such as at the area of land or ice lead, the MCT will detect a large height error. In this case, the OCOG mode will be selected and the SMLE mode will be suspended. When this amplitude difference is small, the SMLE mode will be selected.

Figure 1a,b compare the percentage of different tracker data in different areas of the Arctic. For the areas of land and sea ice margin, due to sudden changes in topography, the OCOG tracker data accounted for most of the data. Furthermore, the SMLE tracker was the primary data source for the ocean and ice sheet zone.

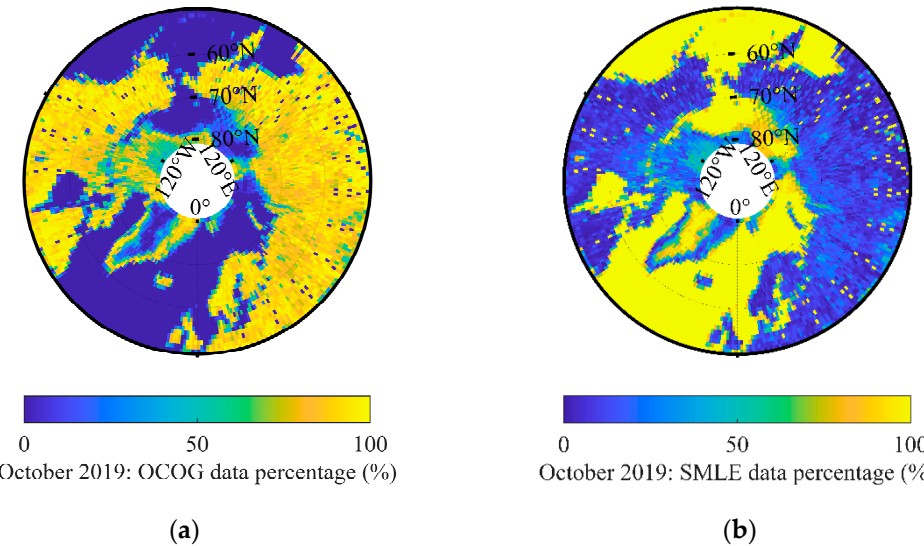

**Figure 1.** The percentage of OCOG and SMLE tracking model data: (**a**) OCOG; (**b**) SMLE.

Figure 2 shows some tracks of the HY-2B altimeter and Sentinel-1 SAR images where the time difference between the two would not exceed four hours. As shown in Figure 2a, the vast majority of the track points of ice leads came from the OCOG tracker (blue points) since the ice leads reflect more energy than the surrounding sea ice. So the amplitude difference between the power spectrum waveform of the ice lead and track window is large. Unlike ice leads, sea ice track points came from the OCOG and SMLE trackers (red points). It may be that some sea ice surfaces are relatively smooth and reflect the signal of the HY-2B altimeter strongly. Overall, these results prove that the detection of ice leads using the HY-2B radar altimeter should include the OCOG tracker data. The L2 data products of the HY-2B altimeter only include data from SMLE, while L1 data products include both SMLW and OCOG. Therefore, in this study, we used the L1B products of the HY-2B radar altimeter in the Arctic from January 2019 to February 2021. In addition, in order to further verify the classification effect of the HY-2B waveform, we used Cryosat-2 L2I data products to compare with the classification results.

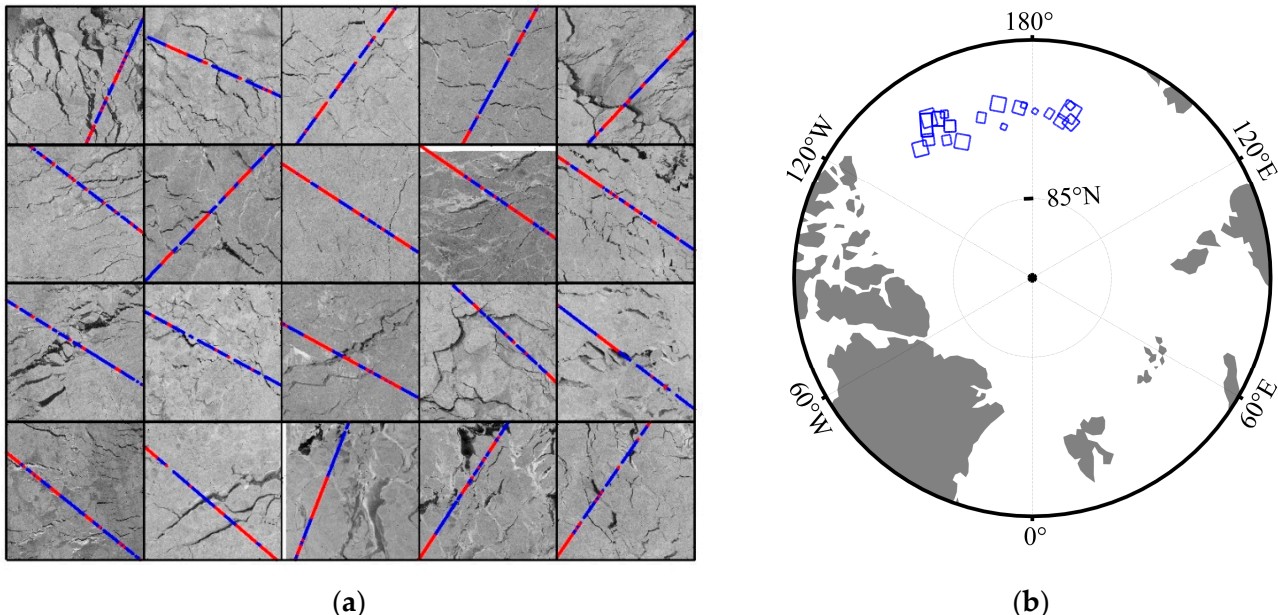

(**a**)         (**b**)

**Figure 2.** Differences in track of HY-2B radar altimeter data for different trackers in sea ice and ice lead: (**a**) the blue points are OCOG. The red points are SMLE. The images are Sentinel-1 SAR image. (**b**) The track of Sentinel-1 SAR image edge in Figure 2a.

### 2.1.2. Sentinel-1

Sentinel-1 contains both A and B satellites and can provide all-weather, day-and-night synthetic aperture radar (SAR) imagery at the C band. Unlike optical satellite images, the SAR images of Sentinel-1 are less affected by clouds and have precise temporal information. There are four radar image acquisition modes: Extra-Wide swath (EW), Stripmap (SM), Interferometric Wide swath (IW), and Wave (WV). In particular, EW mode is primarily used for monitoring areas of sea ice, polar zones, and certain maritime areas. Moreover, EW mode acquires data over a 400 km swath at 20 m × 40 m spatial resolution. Therefore, in this study, we used EW mode SAR images of Sentinel-1 to provide ground truth for the HY-2B radar altimeter waveforms.

The SAR images were pre-processed according to the following steps: (1) radiometric calibration [28], (2) incidence angle correction [29], (3) filter by refined Lee algorithm [30], and (4) K-means image segmentation [31]. To minimize the effects of sea ice variability, all SAR images must be in the same area as the HY-2B, and the time proximity with HY-2B is never more than four hours of time difference.

### 2.1.3. Study Areas and Dates

For HY-2B altimeter waveform type identification, the resulting data consisted of 102 Sentinel-1 SAR images and 74015 HY-2B altimeter waveforms in October 2019. As shown in Figure 3a,b, the selected data traces were mainly distributed in the Beaufort Sea, the Laptev Sea, the Chukchee Sea, and part of the Arctic Central Sea. In this study, the selection of the specific tracks of the HY-2B radar altimeter strongly depended on the Sentinel-1 SAR images. After classification, we also used the trained classifier to map the ice lead fraction for the HY-2B altimeter from January 2019 to February 2021.

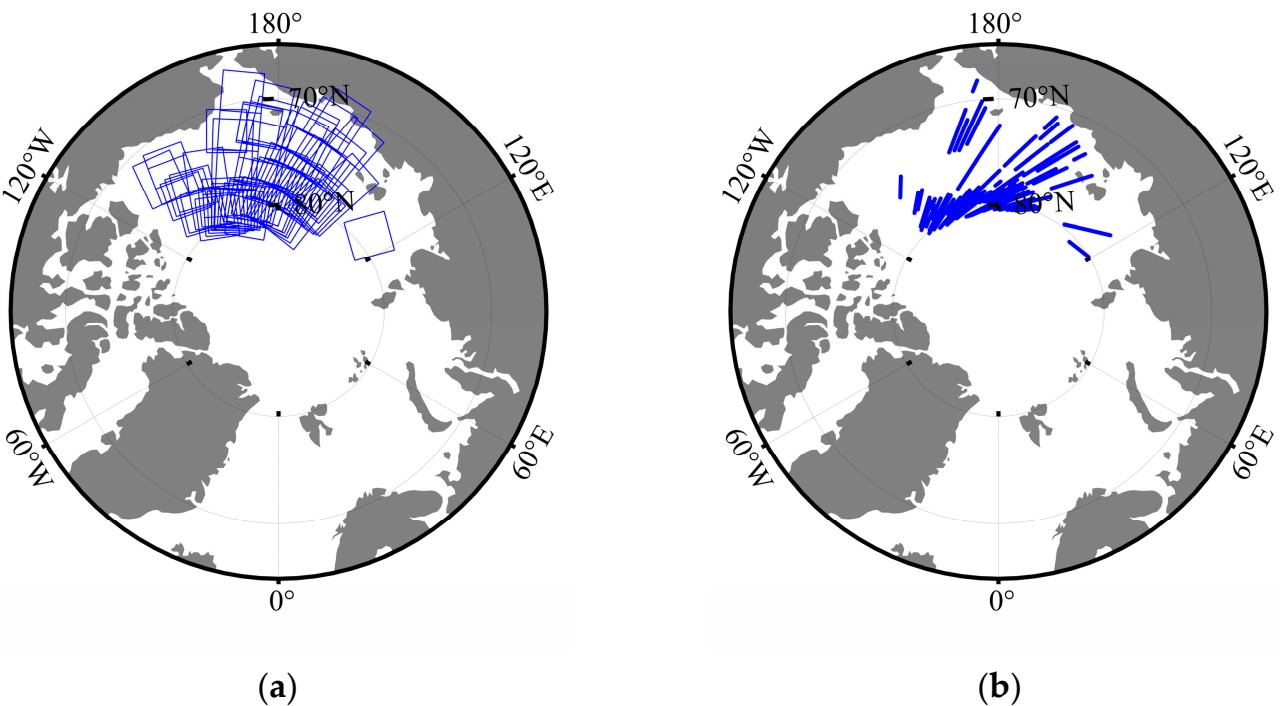

**Figure 3.** Areas to be studied in this paper: (**a**) Sentinel-1 SAR image edge tracks; (**b**) HY-2B radar altimeter tracks. Date: October 2019.

### 2.2. Classifying Parameters and Classifiers

#### 2.2.1. Waveform Parameters

In theory, different surface characteristics within the radar footprint have different scattering characteristics. For instance, a sea ice lead can be considered a specular reflection of microwave (Ku and C bands) due to the calm water surface. Diffuse reflection can occur in open water due to rough surfaces. Furthermore, the surface roughness of sea ice is somewhere in between. Therefore, the waveform of the HY-2B radar altimeter for different surface types has a different shape. As shown in Figure 4, the waveforms of ice leads resemble spike pulses, and waveform energy rises and falls quickly. The waveform energy rises and falls more slowly in open water. Sea ice is in between the lead and the ocean. In order to describe the HY-2B altimeter waveform, we selected the following ten features. These waveform parameters are commonly used to identify leads and ice floes [13,19,20]. Similar to Müller et al. [18], the selection of these waveform parameters is mainly based on the fact that these waveform parameters can represent different types of waveforms and without linear dependence between these parameters.

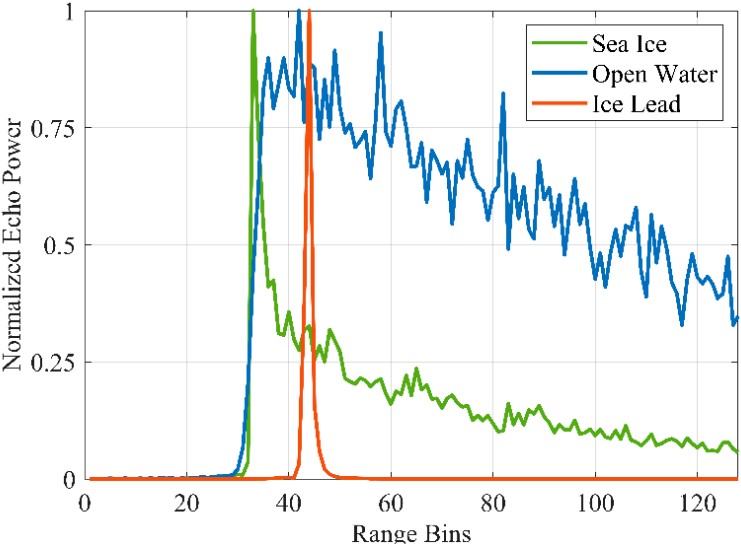

**Figure 4.** HY-2B radar altimeter waveform example of the open water, ice lead, and sea ice.

Pulse Peakiness (PP): PP is the maximum power value ratio to the mean waveform power, which is the same definition as in Jiang et al. [32]. In order to reduce the impact of noise, an improved PP calculation formula, as shown in Equation (1), was used, where $\max(Power)$ is the maximum value in the waveform from bin 21 to bin 88, and $Power(i)$ is the power value in bin $i$. PP is larger on smooth surfaces and smaller on rough surfaces.

$$\text{PP} = \frac{\max(Power)}{\sum_{21}^{108} Power(i)} \times 88 \tag{1}$$

Backscatter coefficient (Sigma0): Sigma0 is calculated by solving the radar equation. Sigma0 is related to surface properties, radar polarization, frequency, and angle of incidence. The backscatter coefficients of different surface types have different values due to differences in surface roughness. In this study, Sigma0 was defined as follows in Equation (2):

$$\text{Sigma0} = AGC + 30 \log_{10}\left[H\left(1 + \frac{H}{R_e}\right)\right] - 30 \log_{10}\left[H\left(1 + \frac{H_0}{R_e}\right)\right] + 10 \log_{10}\left(\frac{A}{2}\right) \tag{2}$$

where AGC is the corrected automatic gain control of radar altimeters that can be used to measure the surface roughness of sea ice and oceans; $H$ is the height of the HY-2B satellite relative to the reference ellipsoid; $H_0$ is a constant with a value of 972,000; $R_e$ is the radius of the Earth; and A is the maximum value of the radar altimeter waveform.

Leading Edge Width (LEW): LEW is defined as the width between 5% and 95% of the leading edge of the waveform's amplitude. Considering that the LEW calculated using the bin is an integer, this cannot provide an accurate LEW value. This article uses Gaussian filtering and linear interpolation to pre-process the waveform.

Trailing Edge width (TEW): TEW is defined as the width between 5% and 95% of the trailing edge of the waveform's amplitude. Similar to LEW, we also performed Gaussian filtering and linear interpolation. As shown in Figure 4, The waveforms of sea ice and open water have a larger TEW, while the TEW of ice leads is smaller.

Pulse Peakiness Left (PPL): PPL is a modified form of PP, defined as the ratio of the maximum power value to the sum of the three bins on the left side of the maximum power, where $i_{max}$ is the bin of the maximum power of the waveform [13,19,33].

$$\text{PPL} = \frac{\max(Power)}{\sum_{i_{max}-3}^{i_{max}-1} Power(i)} \tag{3}$$

Pulse Peakiness Right (PPR): PPR is the right pulse peakiness for the waveform. It uses three bins on the right side of the maximum power to describe the peakiness of the right side of the waveform [13,19,33].

$$\text{PPR} = \frac{\max(Power)}{\sum_{i_{max}+3}^{i_{max}+1} Power(i)} \tag{4}$$

Pulse Peakiness Local (PPLoc): PPLoc uses three bins on the left and right sides of the maximum power to describe the peakiness of the waveform [13].

$$\text{PPloc} = \frac{\max(Power)}{\sum_{i_{max}-3}^{i_{max}+3} Power(i)} \tag{5}$$

Waveform Maximum (MAX): MAX is the maximum power of the waveform.

Kurtosis (Kurt): Kurt is a measure of the peakiness of the power distribution [34]. As shown in Equation (6), Kurt is defined as the ratio of the fourth central moment of the waveform power to the fourth waveform power of standard deviation, where $\mu$ is the mean power of the waveform.

$$\text{Kurtosis} = \frac{E\left[(X-\mu)^4\right]}{\left(E\left[(X-\mu)^2\right]\right)^2} \tag{6}$$

Skewness (Skew): Skew is mainly used to measure the energy distribution's tilt level, which represents the third standardized moment [34]. It can be computed by the third central moment of the distribution by the third power of standard deviation.

$$\text{Skewness} = \frac{E\left[(X-\mu)^3\right]}{\left(E\left[(X-\mu)^2\right]\right)^2} \tag{7}$$

### 2.2.2. Waveform Classifiers

This study assessed ten classifiers, including eight supervised classifiers, one unsupervised classifier, and one thresholding classifier. Before classification, we obtained some waveform samples with echo type labels based on Sentinel-1 SAR images, which are described in Section 3.1. In order to prevent the imbalance of the number of samples from affecting the classification, this paper randomly selected the same number of samples as leads from sea ice and open water, respectively. We then combined these data into a new dataset. This dataset was updated at each training and test in this study for the supervised and thresholding classifiers. Furthermore, at each classification, 3/4 of the dataset was randomly selected for training and the remaining 1/4 for testing.

**Unsupervised classifiers**:

- K-means: K-means is one of the most commonly used unsupervised clustering algorithms [31]. Like [10], the HY-2B radar altimeter data were clustered into 30 clusters. First, $k$ ($k = 30$) samples are randomly selected as the initial clustering centers for the $k$ clusters. Second, the algorithm calculates the distance from all other data to the center of each cluster based on the Euclidean distance. It then finds the nearest cluster center to each sample point and uses this cluster as the cluster of this sample point. Third, it recalculates the new cluster center after each sample point belongs to the corresponding cluster. It then repeats the above steps until it has found $k$ cluster family centers that remain unchanged or change little. Finally, in this paper, according to the waveform characteristics of each cluster, k clusters were artificially divided into three types: sea ice, open water, and ice lead.

**Threshold classifiers**:

- The threshold method is widely used to identify sea ice lead in radar altimeters. Unlike traditional thresholding methods that set thresholds empirically, Wernecke et al. developed a threshold optimization technique that finds the optimal classification threshold based on an iterative process [20]. The technique was also applied in this study. As described in [20], we also used a repeated random cross-validation technique and Nelder–Mead simplex algorithm to minimize the cost function (Equation (8)) to derive and test thresholds $\Theta$:

$$\cos t(\Theta) = w \times \text{False } \overline{\text{Lead}}(\Theta) + \text{False Lead}(\Theta) \tag{8}$$

where $\Theta$ is a vector with the threshold of characteristic parameters, and $w$ is a weighting factor defining how the false classification is minimized. False Lead are observations classified as leads but are actually sea ice or open water. False $\overline{\text{Lead}}$ are observations classified as sea ice or open water but are actually leads. Compared with [20], this study grouped sea ice and open water into the same category and split the data sample into sea ice leads and other categories (sea ice or open water). Because the result of the iteration may be a local optimum rather than a global optimum, this article repeated the calculation 400 times to get the optimal $\Theta$.

**Supervised classifiers**:

- Ensemble learning: Ensemble learning achieves better detection results than any single machine learning model by building and combining multiple learners [35]. The general structure of ensemble learning is to generate a group of basic learners and combine them with some strategies [35,36]. In this study, we used decision trees as the base learner. In addition, this study explored three types of combined methods: Adaptive Boosting (AdaBoost), Random Under Sampling Boosting (RusBoost), and Bagging. AdaBoost is an algorithm that boosts a weak learner into a strong learner by iterating the weights of the base classifier based on misclassified data points, thereby minimizing the loss function. RusBoost is a lifting method using random undersampling, which improves classification performance by random sampling from most categories. Bagging is a parallel integrated learning method that constructs decision trees by random sampling with replacement or bootstrapping from the original data.
- Linear discriminant (LD): LD is a classical linear learning method [36,37]. It tries to project the training samples onto a straight line so that the projection points of the same samples are as close as possible while those of the different samples are as far away as possible. When classifying a new sample point, LD projects it onto the same straight line and then determines the type of the new sample based on the location of the projected point.
- K-Nearest Neighbors (KNN): The KNN classification algorithm finds k number of nearest sample points in the training dataset to the test sample point based on a distance metric [38]. Then, the class with the most occurrences in these k samples was chosen to mark the predicted outcome [39]. In this study, the distance metric was the Euclidean distance.
- Support Vector Machine (SVM): SVM is the most widely used kernel learning algorithm [13,40], which transforms the input samples into a high-dimensional feature space by introducing a kernel function. Then, it finds an optimal classification hyperplane for classification purposes. The classification effect will be different with different kernel functions. In this study, a Gaussian kernel was chosen for the kernel of our SVM classifier.
- Naive Bayes Classifier (NB): The Naive Bayes classifier is based on Bayesian decision theory with the assumption of conditional independence of features [41,42]. The main principles are as follows. First, for a given training data set, the joint probability distribution of the input and output is learned based on the assumption of conditional

independence of the features. Then, based on this model, for an input sample, it computes the category corresponding to the maximum output of the posterior probability based on Bayesian theory.

- Artificial Neural Network (ANN): ANN is a network structure that mimics the biological nervous system and consists of interconnected artificial neurons [36,43,44]. In this study, our neural network structure used three fully connected layers. Each fully connected layer produces ten outputs. The outputs of the first and second layers are processed by the rectified linear unit activation function and passed to the next layer of the network. Additionally, the output of the final fully connected layer is processed by the softmax activation function to obtain the corresponding predicted class labels.

## 3. Results

### 3.1. SAR Image Segmentation and Ground Truth

#### 3.1.1. Sentinel-1 Segmentation

Figure 5 shows an instance of an original Sentinel-1 SAR image and the results of this study after processing based on Section 2.1.2. As shown in Figure 5a, the brightness of different areas of the original SAR image was different due to the different angles of incidence in different areas. Figure 5b is the result of this paper after radiation correction, incidence angle correction, and filtering. Compared with the original SAR image, the brightness of different regions in the processed image is much weaker. Figure 5c illustrates the results of ice lead classification to Figure 5b based on the K-means image segmentation algorithm. Comparing Figures 5a and 5c, we found that both larger and smaller ice leads can be well identified by using the method of this paper.

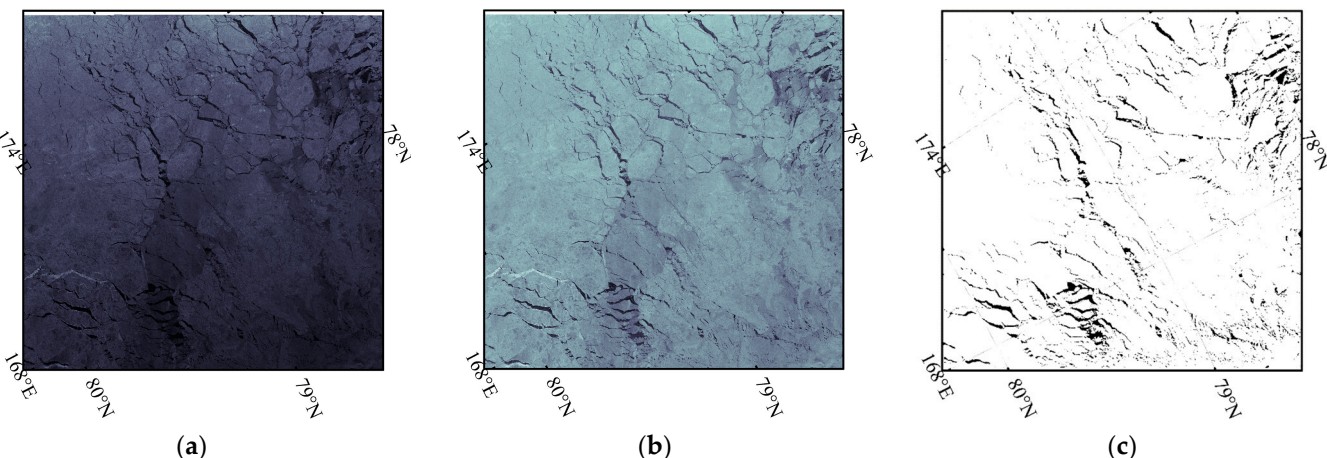

(a)    (b)    (c)

**Figure 5.** Example of Sentinel-1 SAR image processing results: (**a**) the original SAR image; (**b**) the SAR image after radiation correction, incidence angle correction, and filtering; and (**c**) the image after K-means segmentation.

#### 3.1.2. Ground Truth

Because the reflective surface of calm water will dominate the surrounding sea ice return, the effective reflective surface of the altimeter is dramatically reduced to the area of the ice lead when the radar footprint is filled with a mixture of sea ice and ice leads [45]. However, when using satellite imagery to provide ground truth, existing literature determines whether the nadir point coordinates are ice leads. It does not consider the effect of off-nadir point ice leads, which may result in sea ice waveform samples including many lead waveforms. Therefore, the effect of sea ice concentration ($Con_{ice}$) within the radar footprint must be considered when obtaining ground truth from SAR images.

For smooth sea ice and ice lead surfaces, the size of the pulse-limited footprint (effective backscattering area) of the HY-2B radar can be expressed as Equation (9) [46], where $\tau$ is

pulse duration, $c$ is the speed of light, and $h$ is the satellite altitude. When $h = 965$ km and $\tau = 3.125$ ns, we can obtain D $= 1.9$ km for the Ku band of the HY-2B altimeter.

$$D = 2(ch\tau)^{1/2} \tag{9}$$

Over surfaces in which the roughness amplitude distribution exceeds the transmitted pulse width $c\tau$ (about 0.3 m), the footprint of the altimeter becomes [46]:

$$D = 2(ch\tau')^{1/2} \tag{10}$$

where $\tau' = \sqrt{\tau^2 + (16\sigma_h{}^2 \ln 2)/c^2}$; $\sigma_h$ is the rms wave height. This will increase the size where the radar altimeter can accept power. This study used the Sentinel-1 SAR images after processing as described in Section 2.1.2 and acquired 51601 HY-2B radar altimeter waveforms of different concentrations within the effective backscattering area.

Figure 6 shows an example of this study using Sentinel-1 SAR images to count different concentrations of waveforms. For $Con_{ice} < 100\%$, we assume the surface acts as a mirror in this case. Therefore, according to Equation (9), the ratio of the number of sea ice SAR image pixels to the total number of pixels within a 1.9 km diameter of the nadir point is the sea ice concentration of the HY-2B radar altimeter. For $Con_{ice} = 100\%$, different sea ice types have differences in surface roughness. As described in [46,47], the surface roughness of some ice could be larger than the transmitted pulse width $c\tau$. Therefore, we have difficulty accurately determining the effective scattering area of sea ice. To overcome this problem, we only retained sea ice waveforms with a sea ice concentration of 100% within the 10 km (the footprint size of the C band) diameter of the nadir point. This can improve the reliability of the sea ice label. In addition, this also can ensure that the C band footprint is also full of sea ice, thus reducing the impact of the off-nadir ice leads in the process of ionospheric correction using the C band in the measurement of sea ice thickness.

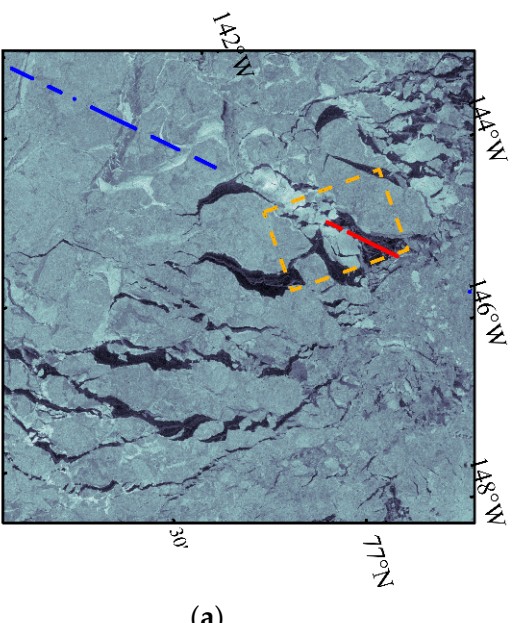
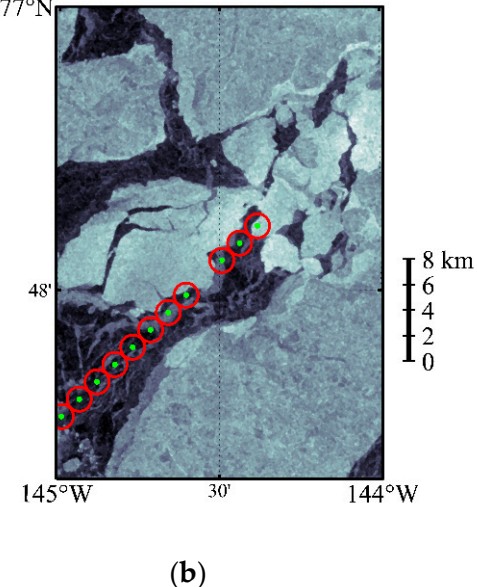

(**a**)          (**b**)

**Figure 6.** Example of ground truth and sea ice concentration for the HY-2B radar altimeter using Sentinel-1 SAR images: (**a**) ground truth example; (**b**) examples of different sea ice concentrations within the radar pulse-limited footprint.

According to the above information, we collected a total of 51601 samples of HY-2B waveforms with different sea ice concentrations. These data could also be used to classify sea ice, ice lead, and open water. Figure 7 shows normalized mean waveforms for different concentration intervals. To obtain an accurate mean waveform power, we flattened the

range bins of the amplitudes of each waveform to 42. As shown in Figure 7, the waveform had larger LEW and TEW when $Con_{ice} = 100\%$. Moreover, with the decrease in sea ice concentration, the LEW and TEW also decreased. When the sea ice concentration was between 90% and 100%, the waveform characteristics of the normalized mean waveform were between the sea ice and the ice lead. When the sea ice concentration was between 80% and 90%, the waveform characteristics were biased toward the characteristics of the ice lead. However, it still had a larger trailing edge width. When the sea ice concentration was below 80%, the waveforms had a more obvious ice lead character, and the normalized mean waveforms did not differ significantly between the different concentration zones.

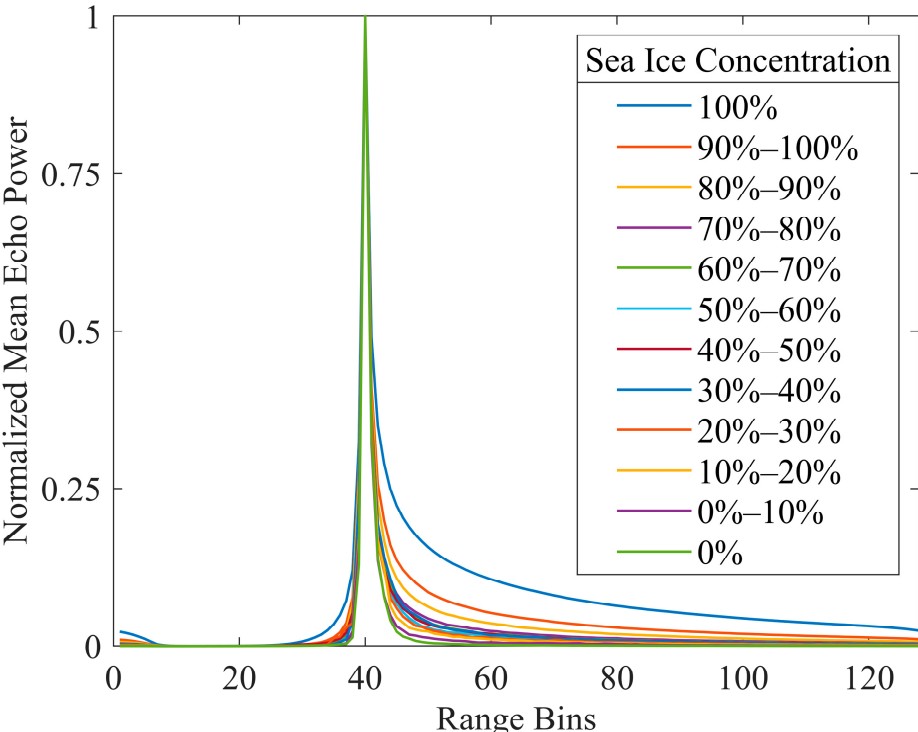

**Figure 7.** The normalized mean waveforms for different sea ice concentrations.

In this article, to ensure accurate labels of ice leads, we removed waveforms with sea ice concentrations in the range of 80–100% and the bin of amplitude less than 20 or more than 100. We set the ground truth for waveforms with $Con_{ice} < 80\%$ as ice leads (2046 samples) and set the waveform of $Con_{ice} = 100\%$ as sea ice (7996 samples). In addition, we obtained 22,414 open water waveforms by artificial matching using Sentinel-1 SAR images (all SAR images include ocean only) which were the same time and position as the HY-2B radar altimeter data.

### 3.2. Analysis of Waveform Parameters

Figure 8 illustrates the cumulative probability distributions of each parameter. In this paper, we calculated ten waveform parameters described in Section 2.2.1. The cumulative probability distribution was used to examine whether the waveform parameters of the radar altimeter are suitable for classification [32]. The greater the separation of the cumulative probability distributions of a parameter of different types, the more suitable that parameter is for the classification.

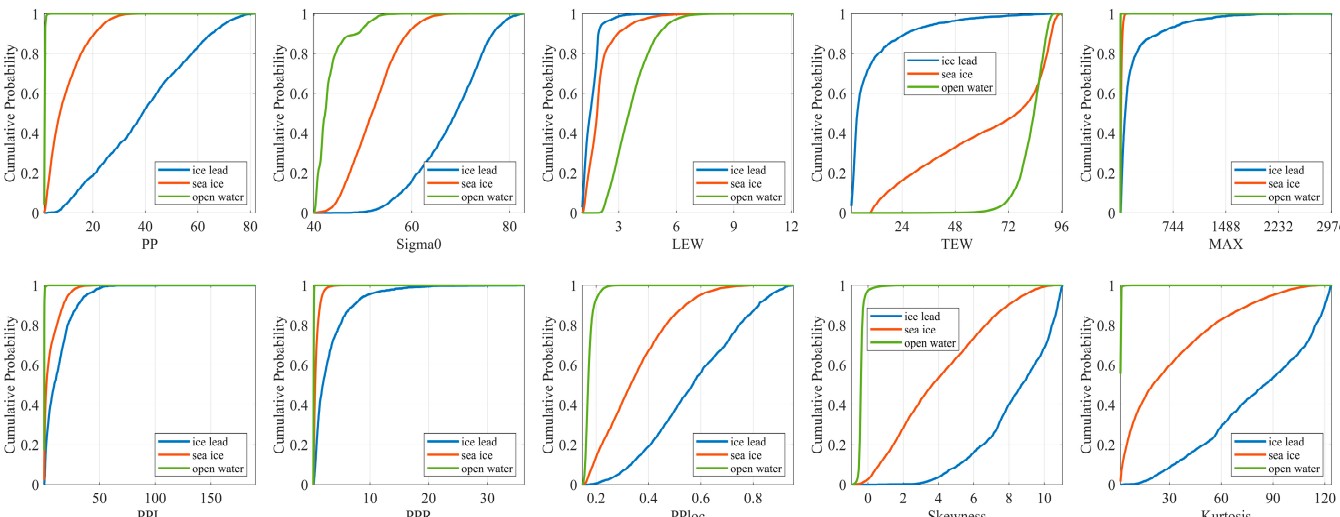

**Figure 8.** The cumulative probability for different waveform characteristic parameters.

As shown in Figure 8, the cumulative probability distributions of PP, Sigma0, TEW, PPloc, Skew, and Kurt in open water, sea ice, and leads possessed significant differences. It was found that the six parameters had good discrimination results for open water, sea ice, and leads. For example, when the PP was at low-value regions, the cumulative probability of the open water almost reached as high as 90%. At the same time, the rates of sea ice and ice leads were significantly lower. The cumulative probability distributions of Sigma0, PPloc, Skew, and Kurt were similar to PP. In addition, the cumulative probability distribution of LEW for sea ice and sea ice leads closely resembled each other, which were small compared to open water. Therefore, LEW can be used to distinguish open water. MAX, PPL, and PPR had similar characteristics in terms of distribution shape, with all three parameters being distinctly different from open water. In addition, the range of values for MAX and PPR was smaller for sea ice and open water compared to sea ice leads. In summary, it was found that all parameters of the HY-2B radar altimeter data used in this study were suitable for identifying their flags.

### 3.3. Classification Performance

This section introduces the classification results of each classifier selected in this study. In order to assess the performance of classification, this paper used the accuracy, positive predictive rate (PPV), Intersection over Union (IoU), and receiver operating characteristic (ROC) curves for evaluation [48].

For classification accuracy, four criteria were computed for the evaluation: overall accuracy ($ACC_{all}$), lead accuracy ($ACC_{lead}$), ice accuracy ($ACC_{ice}$), and open water accuracy ($ACC_{sea}$). $ACC_{all}$ is the ratio of the number of correctly classified samples to the total number of samples. $ACC_{lead}$, $ACC_{ice}$, and $ACC_{sea}$ are the classification accuracy rates of taking corresponding categories as positive and negative categories, respectively.

PPV represents the correct proportion of all positive classes predicted by the model. Similar to accuracy, this paper analyzed three categories of PPV. The greater the classification accuracy and PPV, the better the results. Moreover, the IoU is an evaluation metric used to measure the accuracy of an object detector on a particular dataset [49].

ROC curves are an effective tool for studying the generalization performance of classifiers. The horizontal and vertical coordinates of the ROC curve are false-positive rate (FPR) and true-positive rate (TPR), respectively. FPR represents the proportion of negative tuples incorrectly recognized as positive, and TPR represents the proportion of positive tuples recognized. In addition, the area under the ROC curve (AUC) can quantitatively compare the performance of two classifiers when the ROC curve between the two classifiers crosses.

### 3.3.1. Unsupervised Classifiers

It is known that the return waveform will be a pulsed waveform with a small LEW and TEW when the radar altimeter target type is lead. The rising and falling edge slopes in the waveform of open water are small as the roughness distribution within the radar footprint is greater than the altimeter transmitted pulse width. The scattering characteristics of sea ice are between the open water and the lead, where the slope of the falling edge is larger than the open water, and the width of the rising edge is larger than the lead.

For the K-means clustering classifier, we clustered the HY-2B radar altimeter data into 30 clusters. Figure 9 shows the result of HY-2B radar waveform clusters after K-means clustering. In this study, we classified clusters 3, 4, 7, 9, 11, 14, 17, 19, 21, 25, 27, and 30 as sea ice, clusters 1, 10, 12, 15, and 20 as open water, and clusters 2, 5-6, 8, 13, 16, 18, 22-24, 26, and 28-29 as ice leads based on the waveform characteristics described above. The overall accuracy of K-means was 83.24%, and the accuracy of sea ice, open water, and ice leads were 83.32%, 92.44%, and 90.71%, respectively.

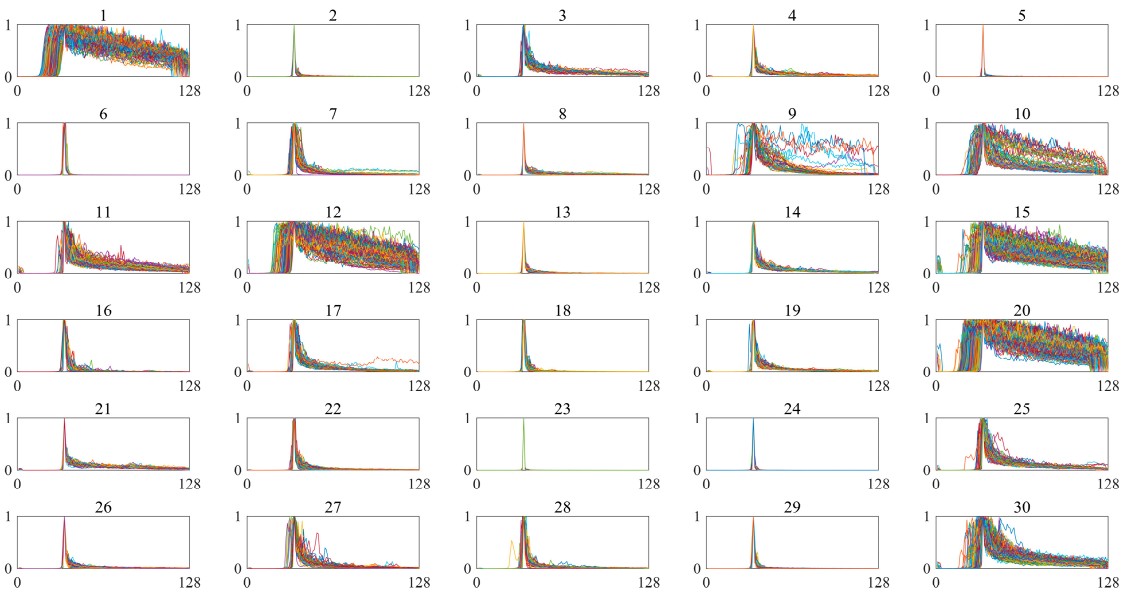

**Figure 9.** HY-2B altimeter waveform clusters (k = 30) after K-means clustering showing segmented waveforms.

### 3.3.2. Threshold Classifiers

As described in Section 2.2.2, the threshold classifier's cost function (see Equation (8)) has a weighting factor. The selection of different weighting factors has different meanings [19,20]. When $w = 1$, the complete false classification is minimized. When $w < 1$, this indicates that one might be interested in reducing the amount of sea ice or open water incorrectly detected as lead. It could obtain a more conservative result in lead detection, while the amount of lead detected as sea ice or open water will increase. Furthermore, when $w > 1$, the minimization strategy of the cost function is opposite to when $w < 1$. This study applied eight weighting factors (0.05, 0.25, 0.5, 0.75, 1, 2, 5, and 10) to capture the performance development.

Table 1 shows the statistical average of the optimized threshold values and the lead accuracy ($ACC_{lead}$) of different weighting factors. The produced lead accuracies were generally high, with most weighting factors producing more than 85% lead accuracy. As shown in Table 1, the lead accuracy rates corresponding to different weight factors were great, except when $w = 0.05$. For $w = 0.5$ and $w = 0.05$, the threshold classifier had the highest and lowest lead accuracy rate (90.23%), respectively. Moreover, for $w = 0.75$, the accuracy was also more significant than 90%.

**Table 1.** The optimized thresholds and their corresponding weight and lead accuracy.

| w | PP | Sigma0 | LEW | TEW | MAX | PPL | PPR | PPLoc | Skew | Kurto | ACC$_{lead}$ |
|---|---|---|---|---|---|---|---|---|---|---|---|
| 0.05 | 20.8236 | 64.8120 | 7.5238 | 64.0173 | 24.1670 | 0.3027 | 0.2711 | 0.3895 | 6.7238 | 50.3477 | 85.98% |
| 0.25 | 20.8299 | 59.9932 | 5.7643 | 34.6195 | 14.8116 | 0.2306 | 0.2114 | 0.3560 | 6.0659 | 44.3631 | 89.26% |
| 0.5 | 16.7404 | 58.5302 | 5.8184 | 18.3020 | 14.5760 | 0.4292 | 0.1516 | 0.3371 | 5.7580 | 40.7821 | 90.23% |
| 0.75 | 16.5950 | 57.7662 | 6.2133 | 19.2390 | 13.8309 | 0.4460 | 0.1163 | 0.3445 | 5.7626 | 41.0216 | 90.14% |
| 1 | 16.6628 | 56.8634 | 7.0824 | 19.4076 | 13.5906 | 0.5679 | 0.0911 | 0.3491 | 5.7889 | 41.3066 | 89.95% |
| 2 | 16.4168 | 56.9398 | 7.1996 | 23.0575 | 13.1400 | 0.6833 | 0.1512 | 0.3496 | 5.6865 | 40.1072 | 89.76% |
| 5 | 14.5733 | 56.0381 | 8.9725 | 29.5261 | 13.1718 | 0.6116 | 0.3181 | 0.3383 | 5.2954 | 37.1081 | 89.27% |
| 10 | 14.8781 | 55.8707 | 8.5477 | 28.8143 | 13.1375 | 0.6176 | 0.2671 | 0.3435 | 5.3398 | 37.9540 | 89.13% |

When the $w$ increased from 0.05 to 0.5, this study found a gradual increase in the lead accuracy rate. When the $w$ changed from 0.5 to 10, the lead accuracy rate was the opposite. Furthermore, when $w = 0.5$, the LEW, TEW, PPL, PPR, and PPLoc had smaller values, which was closer to the sharp pulse waveform characteristics of the lead. A comparative analysis of the threshold classifier's accuracy and other classifiers is discussed in Section 3.3.3.

### 3.3.3. Supervised Classifiers

Figure 10 illustrates the ROC curves and area under the supervised classifier's ROC curve (AUC). It is apparent from Figure 10 that all of the supervised classifiers produced an excellent performance. For example, the ROC curves of lead and ice for all supervised classifications were as close as possible to the upper left corner of the coordinates. Apart from this, the AUS values of ice and lead were all >0.94. Compared with [13], the results of ROC curves in this study had greater AUC. This is due to the sea ice concentration being considered in the ground truth of lead and ice, reducing the impact of incorrect labels. Furthermore, the inclusion of open water samples was more easily detected in this study.

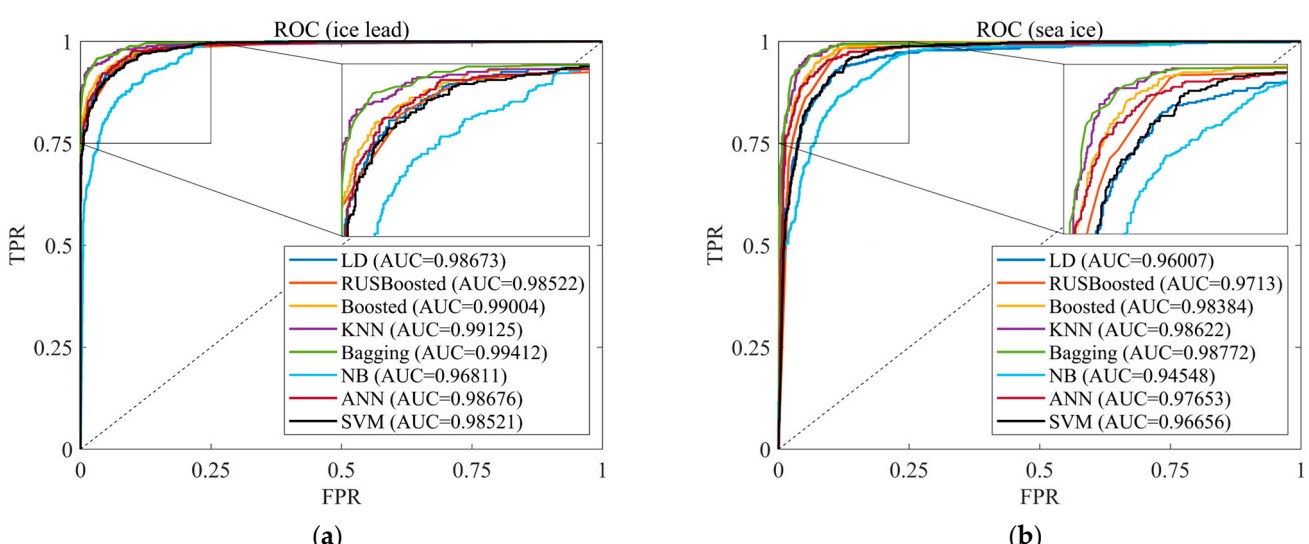

**Figure 10.** ROC curve and AUC of supervised classifier: (**a**) lead; (**b**) sea ice.

From Figure 10, it can be seen that Bagging and KNN classifiers showed very similar results throughout the ROC curve and AUC. In addition, the performance of these two classifiers was more outstanding than other supervised classifiers. The performance of the Bagging classifier in this study was similar to those of [13]. The AUC of bagging and KNN classifiers for ice leads were 0.99412 and 0.99125, respectively. Furthermore, the AUC of bagging and KNN classifiers for sea ice were 0.98772 and 0.98622, respectively. The NB classifier performed worse than other supervised classifiers and produced lower AUC.

Given this preliminary analysis of the supervised learning classifiers, this study has found that the Bagging and KNN classifiers' performance is better than other supervised learning classifiers.

We now turn to the experimental result on accuracy and positive predictive value. The classification performances of all supervised and K-means classifiers are given in Table 2 and Figure 11. All of the supervised classifiers produced more than 85% overall accuracy, while the unsupervised classifier (K-means) had 83.24% overall accuracy. In addition, the $ACC_{sea}$, $ACC_{lead}$, and $ACC_{ice}$ of supervised classifiers were larger than the K-means classifier. Therefore, it seems that the overall performance of the supervised classifiers was better than the K-means classifier.

**Table 2.** Results of classification performance metrics for supervised and unsupervised learners.

| Classifier | RUS Boosted | Boosted | Bagging | LD | KNN | SVM | NB | ANN | K-Means |
|---|---|---|---|---|---|---|---|---|---|
| $ACC_{overall}$ | 92.33% | 93.20% | 95.69% | 90.59% | 94.52% | 90.29% | 87.00% | 93.09% | 83.24% |
| $ACC_{lead}$ | 93.79% | 94.34% | 96.56% | 93.95% | 96.22% | 93.59% | 89.46% | 94.71% | 90.71% |
| $ACC_{ice}$ | 92.45% | 93.18% | 95.54% | 90.66% | 94.88% | 91.08% | 87.21% | 93.52% | 83.32% |
| $ACC_{sea}$ | 98.64% | 98.84% | 98.98% | 96.65% | 98.63% | 97.45% | 97.80% | 98.78% | 92.44% |
| $TPR_{lead}$ | 0.8976 | 0.8759 | 0.9504 | 0.8984 | 0.9488 | 0.8933 | 0.8133 | 0.9197 | 0.8524 |
| $TPR_{ice}$ | 0.9033 | 0.9381 | 0.9329 | 0.8259 | 0.9062 | 0.8497 | 0.8233 | 0.9001 | 0.6471 |
| $TPR_{sea}$ | 0.9754 | 0.9814 | 0.9829 | 0.9946 | 0.9910 | 0.9887 | 0.9797 | 0.9854 | 0.9976 |
| $FPR_{lead}$ | 0.0419 | 0.0228 | 0.0268 | 0.0400 | 0.0311 | 0.0428 | 0.0655 | 0.0392 | 0.0655 |
| $FPR_{ice}$ | 0.0633 | 0.0713 | 0.0333 | 0.0531 | 0.0299 | 0.0587 | 0.1035 | 0.0473 | 0.0738 |
| $FPR_{sea}$ | 0.0081 | 0.0081 | 0.0068 | 0.0475 | 0.0160 | 0.0326 | 0.0228 | 0.0110 | 0.1122 |
| $PPV_{lead}$ | 89.76% | 87.59% | 95.04% | 89.84% | 94.88% | 89.33% | 81.33% | 91.97% | 85.24% |
| $PPV_{ice}$ | 90.03% | 93.81% | 93.29% | 82.59% | 90.62% | 84.97% | 82.33% | 90.01% | 64.71% |
| $PPV_{sea}$ | 97.54% | 98.14% | 98.29% | 99.46% | 99.10% | 98.87% | 97.97% | 98.54% | 99.76% |
| $IoU_{lead}$ | 0.8337 | 0.8428 | 0.9055 | 0.8309 | 0.8907 | 0.8366 | 0.7144 | 0.8459 | 0.7537 |

**Figure 11.** Classification accuracy of different classifiers. The blue line is overall accuracy. The green line is ocean accuracy. The red line is sea ice accuracy. The purple line is ice lead accuracy.

Comparing all supervised classifiers selected in this study, the Bagging classifier of ensemble learning had the highest $ACC_{all}$ (95.69%), $ACC_{sea}$ (98.98%), $ACC_{lead}$ (96.56%), and $ACC_{ice}$ (95.54%). The KNN classifier has a similar performance to the Bagging classifier. Additionally, the LD classifier has the lowest performance for the supervised learning method. As described in Section 3.3.2 and Figure 11, the $ACC_{lead}$ of the threshold and K-means classifiers were 90.23% and 90.71%, respectively. Compared with the Bagging classifier ($ACC_{lead}$ = 96.56%), the threshold and K-means classifiers had weaker classification performances. Overall, these results show that the Bagging classifier performs best for classifying the HY-2B altimeter radar data.

Interestingly, this study found that both the supervised and K-means classifiers had $ACC_{sea} > ACC_{lead} > ACC_{ice}$. It may be that compared with sea ice and ice leads, open water has obvious waveform characteristics. As a result, open water is easier to identify. In addition, the waveform characteristic distributions of sea ice were between open water and ice lead. For example, the distributions of PP, Sigma0, and Skewness (see Figure 9) showed that the sea ice is embedded between leads and open water, which receives both open water and lead disturbance during classification. Therefore, the sea ice has the lowest corresponding accuracy. Due to the existence of sea ice, ice leads and open water have little interaction in the classification process.

In addition to classification accuracy, this study also used PPV, TPR, and FPR to analyze the performance of the classifier. As described in Table 2, the Bagging classifier had higher $PPV_{lead}$, $PPV_{ice}$, $PPV_{sea}$, and $IoU_{lead}$, while the KNN classifier had the highest $PPV_{sea}$. Since the performance evaluation indexes of Bagging and KNN classifiers were close, it was necessary to study their differences further. Under the assumption that the ice lead was positive, this paper examined the differences between the two classifiers using the McNemar mid-p test technique [36]. After 100 repeated experiments, the mean mid-p-value was 0.4212, which indicates that both Bagging and KNN classifiers have excellent performance [50]. However, due to the accuracy, $PPV_{lead}$, and $PPV_{ice}$ of the Bagging being higher than the KNN, by contrast, we think the overall performance of the Bagging classifier was better than the KNN classifier. Moreover, compared with other classifiers, the Bagging classifier had the largest $TPR_{lead}$ and smaller $FPR_{lead}$. Additionally, the TPR and FPR of sea ice and open water also showed excellent performance. These results again validated the performance of the Bagging classifier.

As described above, the Bagging classifier performed better than other methods. In order to further analyze the performance of Bagging, 1539 samples were randomly selected from the HY-2B radar altimeter data, and the number of samples for the three categories was equal. We then used the trained Bagging classifier to test classification. Figure 12 shows the confusion matrix of the test result. The accuracy of open water was greater than sea ice and leads. Moreover, sea ice had the lowest accuracy.

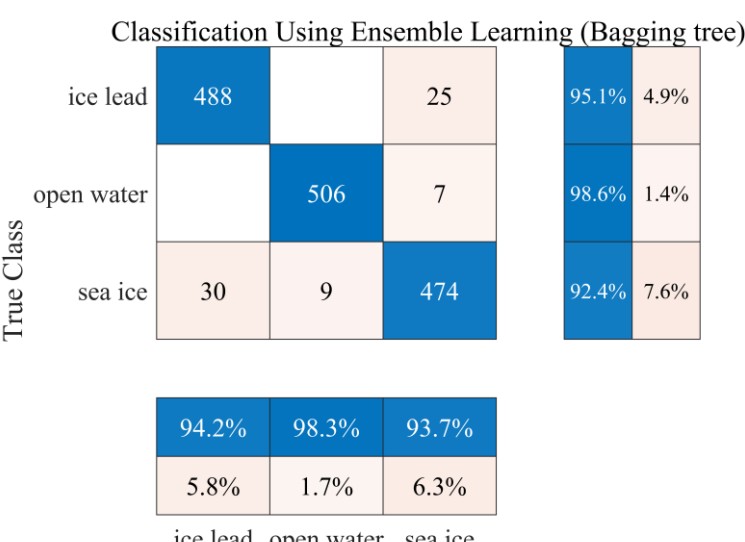

**Figure 12.** Confusion matrix of Bagging classifier.

Interestingly, all the samples incorrectly identified as open water or leads were sea ice. Additionally, ice leads and open water will not be incorrectly identified as each other. Therefore, sea ice will be disturbed by both open water and leads during the classification process. The performance was evident in the fact that sea ice had a lower classification accuracy rate (see Table 2 and Figure 12). From Figure 12, we can know that the percentages

of samples identified as incorrect for sea ice, open water, and ice leads were 7.6%, 1.4%, and 4.9%, respectively. This robust result can meet the need to identify more accurate ice leads during sea ice freeboard and thickness inversions. Therefore, this study used the trained Bagging classifier to map the monthly fraction of leads of the HY-2B altimeter and compared it with the CraySat-2 altimeter in the next section.

*3.4. Comparison with CryoSat-2*

Existing research often uses CryoSat-2 synthetic aperture radar altimeter data to retrieve sea ice freeboard and thickness. The present study was designed to determine the classifier suitable for the HY-2B radar altimeter, which is a critical step of sea ice thickness retrieval. Therefore, it was necessary to compare the classification results of the Bagging classifier obtained in the previous section with CryoSat-2.

In order to investigate the performance of the classifier in practical application, this study classified the HY-2B radar altimeter data every month using the trained Bagging classifier described in the previous section. Then these classified results were mapped to the sea ice concentration, sea ice fraction, and lead fraction in a $0.5^\circ \times 0.5^\circ$ (latitude $\times$ longitude) grid cell. The sea ice concentration is the ratio of the sum of sea ice and leads to the number of all data, assuming sea ice and leads are the same types. Additionally, the ice fraction and lead fraction are the proportion of sea ice and ice leads to the total sample, respectively. Therefore, the sum of the sea ice fraction and the lead fraction equals the sea ice concentration. Among them, the lead fraction has been used to investigate the performance of lead detectors [20]. The lead fraction not only can obtain the spatial distribution characteristics of the lead but also predict the changes in sea ice.

Figure 13 shows the sea ice concentration, ice fraction, and lead fraction obtained in April and October 2019 in this article using the HY-2B radar altimeter data and trained Bagging classifier. From Figure 13, the sea ice and leads had reasonable spatial distribution. For instance, the sea ice concentrations derived from the results from the Bagging classifier had a transparent distribution and shape in the sea ice areas. The sea ice concentration was relatively low for the sea ice margin zone. In contrast, the sea ice concentration was about 100% for the interior of the sea ice area. In addition, as can be seen from the lead fraction, the leads were predominantly located in the sea ice margin zone. Especially in the Greenland Sea, there are a lot of ice leads, which illustrates the dramatic changes in the sea ice. This result is consistent with the Greenland Sea's many ice leads. As shown in the sea ice fraction (yellow zone), this paper also found fewer leads inside the sea ice area. These reasonable distributions of ice and leads generally suggest that the Bagging classifier can be applied to all the HY-2B radar altimeter data in April and October 2019.

Figure 14 shows the lead fraction in April and October 2019 based on the CryoSat-2 altimeter and the bias from the lead fraction in this study. The lead fraction of the CryoSat-2 was calculated based on the L2I product, which includes a flag about the surface type [51]. Furthermore, the lack of estimates in the north of Canada (see Figure 14a) was caused by the use of the SARIn mode in the Wingham Box [51]. When calculating the ice fraction, this study removed the data of undefined categories in CryoSat-2 L2I products. As shown in Figure 14, the distribution characteristics of leads based on CryoSat-2 were consistent with the results of HY-2B in this study. For example, the CryoSat-2 lead fraction similarly indicated the presence of many leads in the Greenland Sea in April and October 2019. In order to quantitatively analyze the difference between the HY-2B and Cryosat-2, the lead fraction bias was obtained by subtracting CryoSat-2 from HY-2B, where the grid cell with a sea ice concentration of zero was removed. From Figure 14, this paper found that the lead fraction bias of each grid cell between HY-2B and CryoSat-2 was relatively small. Figure 15 shows the approximate Gaussian distribution of the bias of the lead fraction. The mean biases were −8.273% (Oct) and −7.3509% (Apr), and the standard deviations (std) were 27.3438% (Oct) and 24.6361% (Apr). Although the average value of the bias was small, the standard deviation was relatively large. These results, therefore, need to be interpreted

with caution. Additionally, it may be that the system error is due to different HY-2B and CryoSat-2 tracks.

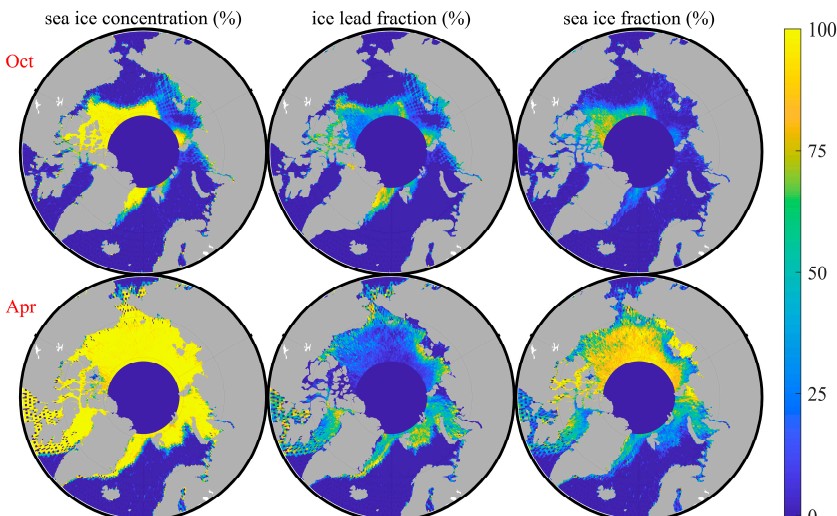

**Figure 13.** The sea ice concentration, sea ice fraction, and lead fraction based on the classification of the Bagging classifier from April and October 2019.

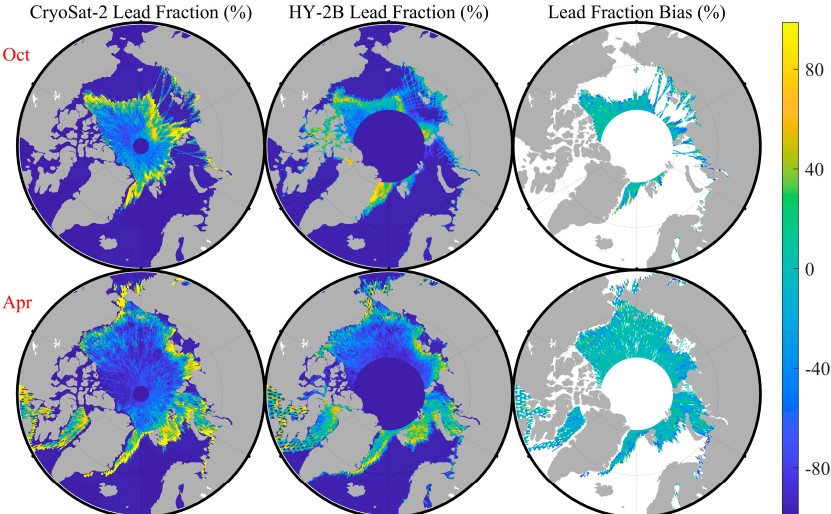

**Figure 14.** The lead fraction base on the CryoSat-2 and HY-2B from April and October 2019 and the lead fraction bias between them.

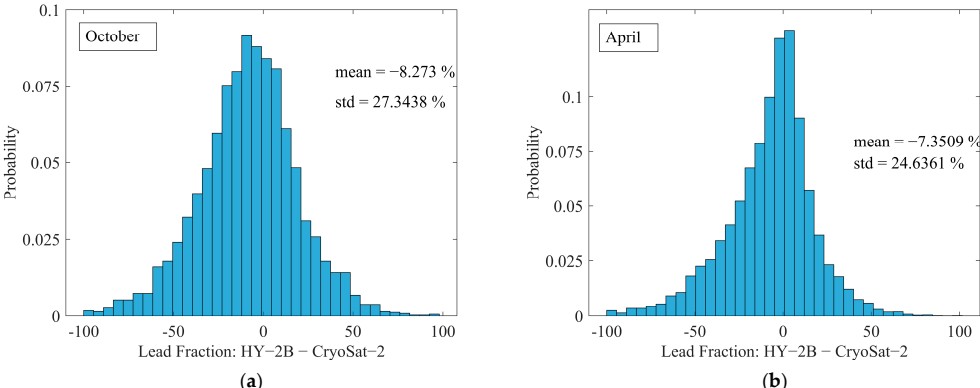

**Figure 15.** Histograms of bias for April and October 2019: (**a**) April; (**b**) October.

All of the above analyses were for April and October 2019. In order to examine the applicability of the trained Bagging classifier to other times, this paper studied the HY-2B radar altimeter data from January 2019 to March 2021. Figure 16 illustrates the monthly lead fraction based on the HY-2B altimeter data in 2019. The results of other months can be found in Appendix A.

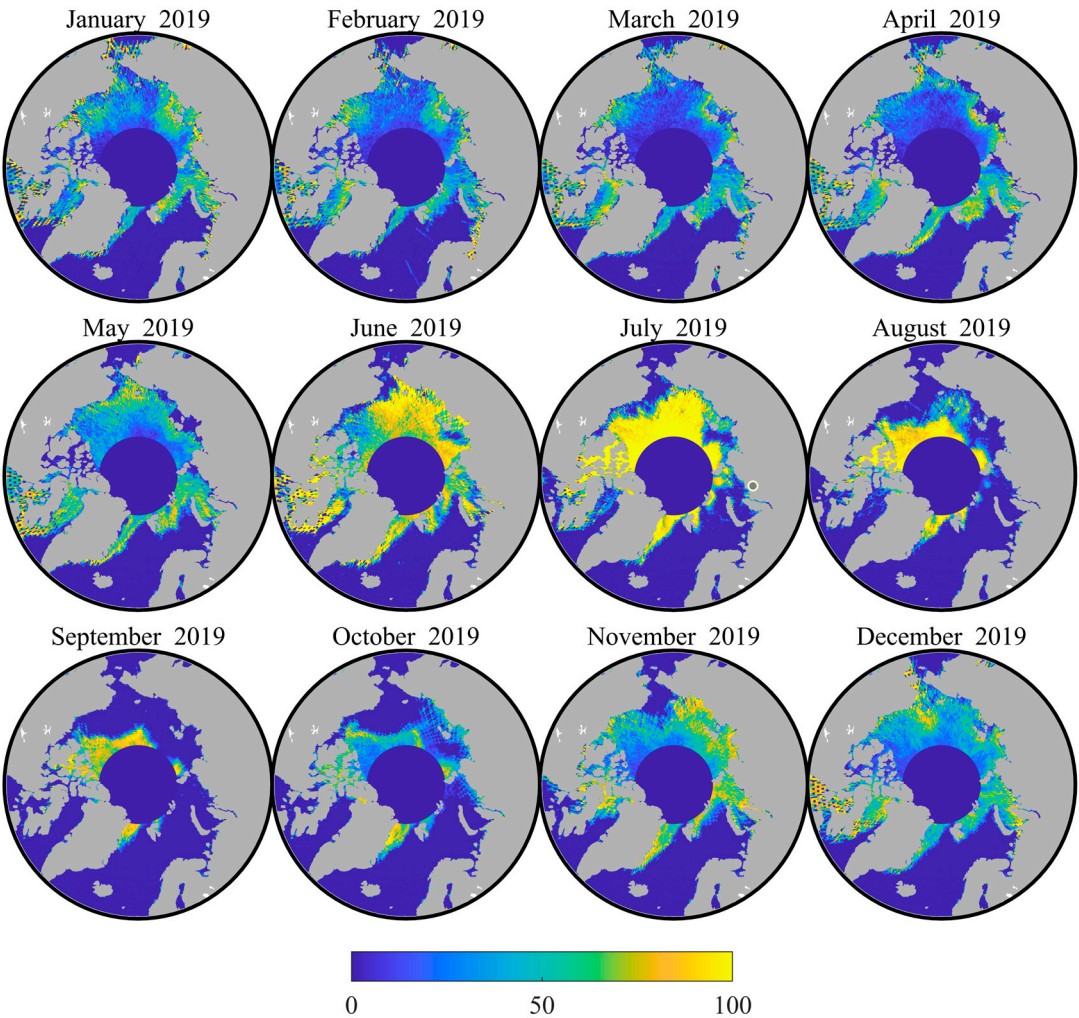

**Figure 16.** Lead fractions from 2019 based on the HY-2B radar altimeter data and trained Bagging classifier.

From January 2019 to February 2019, there was a slight decrease in the lead (such as the Beaufort Sea), which indicates some leads were frozen during this period. Figure 16 presents the melting and breaking of sea ice during the spring of 2019, increasing the number of leads. The ice lead had increased dramatically for the summer season compared with the previous season. Meanwhile, the sea ice area decreased rapidly. After September, the temperature in the Arctic began to decrease again. The lead fraction and ice extent in various regions gradually fell and increased, respectively. In summary, these results show that the variability in sea ice distribution exhibited by the lead fraction based on the HY-2B altimeter is consistent with the actual sea ice situation.

Table 3 provides the monthly bias and standard deviations of the HY-2B and CryoSat-2 monthly lead fraction. According to Table 3, the mean bias and mean std were −9.7877% and 24.5650%, respectively. Additionally, the absolute maximum and minimum biases were 17.9815% (January 2019) and 0.5766% (August 2021), respectively. In addition, the study found that the lead fraction had a large bias in winter, but it was the opposite in summer.

**Table 3.** The mean biases and standard deviations between HY-2B and CryoSat-2 from January 2019 to December 2020.

| Month | 2019 | | 2020 | |
|---|---|---|---|---|
| | Mean (%) | Std (%) | Mean (%) | Std (%) |
| January | −17.9815 | 25.9329 | −15.456 | 23.3506 |
| February | −14.0643 | 25.6564 | −13.0901 | 24.1542 |
| March | −10.3801 | 23.7364 | −10.3014 | 23.1645 |
| April | −7.3509 | 22.5721 | −10.5358 | 23.8801 |
| May | −8.5689 | 24.8613 | −11.3366 | 25.859 |
| June | −9.4941 | 26.3777 | −7.5229 | 21.8657 |
| July | −2.3509 | 22.0199 | −1.8766 | 21.8641 |
| August | −1.6363 | 24.4449 | −0.5766 | 21.4021 |
| September | −6.2395 | 25.2744 | −4.8066 | 26.9569 |
| October | −8.273 | 27.3438 | −8.837 | 26.6601 |
| November | −13.3645 | 25.3374 | −15.0677 | 25.6767 |
| December | −17.8141 | 24.9352 | −17.98 | 26.2347 |

In summary, the monthly lead fraction obtained based on the Bagging classifier showed a reasonable spatial distribution of ice leads and the consistency of the results and CryoSat-2 L2I products. These results enhance our confidence in the HY-2B lead classifier. Therefore, we know that the Bagging classifier's classification results can be used for subsequent inversion of sea ice information.

## 4. Discussion and Conclusions

This paper aimed to assess different classification methods for waveform type detection in the Arctic Ocean using HY-2B radar altimeter data. This study compared eight supervised classifiers, an unsupervised classifier, and an improved threshold classifier. We found that the Bagging tree classifier in ensemble learning performed best. As described in Table 2 and Figure 10, the Bagging classifier had the best ACC, AUC, and PPV compared to other classifiers. To further verify the performance of the Bagging classifier, this study discussed the confusion matrix of the Bagging classifier (as shown in Figure 12). We found that for randomly selected test sets, the selected Bagging classifier also had excellent performance. Moreover, in addition to reasonable spatial distribution, the HY-2B lead fraction based on the Bagging classifier was consistent with the lead fraction based on the CryoSat-2 L2I product. These results show that the Bagging classifier obtained in this study has robust performance and can provide accurate waveform ground truth value for the research of HY-2B radar altimeter calculating sea ice thickness.

Moreover, this study used Sentinel-1 SAR images to statistically analyze the normalized mean waveforms for different sea ice concentrations within the effective backscattering area of the HY-2B radar altimeter. We found that smaller ice leads also may dominate the surrounding sea ice return. As shown in Figure 7, with the decrease in sea ice concentration, the HY-2B radar altimeter waveform characteristics gradually changed from sea ice to ice leads. The HY-2B altimeter waveform had prominent ice lead characteristics when the sea ice concentration is below 80%. Unlike the existing research [13,18,21], this study provides the ground truth value for HY-2B altimeter waveforms based on the sea ice concentration within the pulse-limited footprint, which can provide a more robust data label for the classifier.

By analyzing the different tracker modes of the HY-2B altimeter, this study found that the track points of the OCOG package were mainly distributed in land and sea ice areas. In contrast, the track points of the SMLE package were primarily distributed in the ocean and ice sheets. In addition, in the data set of this paper, 91.54% of the ice lead waveforms came from the OCOG package. As shown in Figure 2, when ice leads appear at the track point, the tracker mode of waveform data almost adopts the OCOG mode. This phenomenon is consistent with the hardware design of the HY-2B radar altimeter [26,27]. Due to the

smooth surface of the ice lead (specular reflection), the difference in the leading edge between the envelope of the power spectrum and the track window (a rectangle window) is significant, which could lead to the OCOG tracker being selected and the SMLE tracker being suspended [26]. Therefore, compared with Dong et al. and Zhang et al. [24,25], the results of waveform classification in this paper are more robust and credible due to using the HY-2B altimeter L1B products, which contain data from both the OCOG tracker and SMLE tracker.

Sea ice is one of the most sensitive environmental factors in the climate system since it influences the exchange of heat and mass between the atmosphere and ocean and the surface radiation balance [1,52]. Large-scale changes in sea ice can lead to changes in the global distribution of heat and cold sources, which can have important implications for regional and global warming. Therefore, monitoring sea ice properties is particularly important for studying global climate change. Sea ice thickness, as the third dimension of sea ice, can be combined with sea ice density to calculate sea ice volume to better understand changes in sea ice [24]. Thus, researchers have shown an increased interest in sea ice thickness. Among many methods available today [12,53–55], the satellite altimeter is the only one that can monitor sea ice thickness on a hemispheric scale. In addition, identifying ice leads is the most critical part of the altimeter inversion of sea ice thickness. Compared with the majority of the current literature that calculates sea ice thickness based on the CryoSat-2 radar altimeter [12,15,16], the research of HY-2B radar altimeter in sea ice thickness is not perfect. In this study, we assessed different classification methods for waveform type detection in the Arctic Ocean based on the HY-2B altimeter, which can provide an essential reference for future studies of sea ice freeboard and sea ice thickness.

The major limitation of this study is that only October data from parts of the Arctic were used for classification. For example, since only a tiny number of Sentinel-1 images can be used to provide the ground truth for the HY-2B altimeter in the Greenland Sea (it is challenging to find data for the same region at the same time), this study did not use the data of this area for classification research. In future work, we will use data that include additional years and months to assess the performance of the HY-2B altimeter regarding waveform classification. Moreover, the Antarctic region also contains a large amount of sea ice. In the future, we will carry out the waveform classification research of the HY-2B altimeter in the Antarctic region. Additionally, we will use these better classifiers and the HY-2B radar altimeter to retrieve and evaluate the sea ice thickness in the Arctic and Antarctic.

Moreover, The CryoSat-2 L2I class contains many unknown categories of data whose waveform characteristics are mainly between sea ice and inter-ice channels. For example, as shown in Figure 14, removing these waveforms made the CryoSat-2 lead fraction show more dramatic changes in the spatial distribution of sea ice compared with the HY-2B radar altimeter. In addition, conventional pulse-limited altimeters such as HY-2B have a typical footprint of 2–10 km over sea ice. Unlike HY-2B, the SAR technology and the Doppler post-processing offer CryoSat-2 radar altimeter an along-track footprint of ~300 m. These reduce the impact of off-nadir ice leads and snow on the waveform of CryoSat-2 [56]. These differences increase the bias between the HY-2B and CryoSat-2 lead fractions. Furthermore, this will increase the uncertainty of sea ice thickness based on conventional altimeters such as HY-2B. Fortunately, some studies have shown that the sea ice thickness deviation between the traditional altimeter and the CryoSat-2 SAR altimeter can be improved by correction [56–59]. In the future, we will study the influence of HY-2B sea ice identification results on sea ice thickness deviation.

**Author Contributions:** Conceptualization, W.Z., M.J. and K.X.; methodology, W.Z., M.J. and K.X.; software, W.Z.; formal analysis, W.Z. and M.J.; investigation, W.Z.; data curation, M.J. and Y.J.; writing—original draft preparation, W.Z., M.J. and K.X.; writing—review and editing, W.Z., M.J. and K.X.; project administration, M.J. and K.X.; funding acquisition, M.J. and K.X. All authors have read and agreed to the published version of the manuscript.

**Funding:** This research was funded by the National Natural Science Foundation of China (Grant No. 41906199), the Foundation of Key Laboratory of Space Ocean Remote Sensing and Application, MNR (No. E0C01kA340), and the Youth Innovation Project of National Space Science Center of Chinese Academy of Sciences (No. EOPD40012S).

**Data Availability Statement:** The HY-2B radar altimeter data were downloaded from the National Satellite Ocean Application Service (NSOAS), available online: https://www.nsoas.org.cn (accessed on 29 November 2021). The Sentinel-1 SAR image data were downloaded from the Alaska Satellite Facility (ASF), available online: https://search.asf.alaska.edu/#/ (accessed on 21 January 2022). The CryoSat-2 radar altimeter data were downloaded from the European Space Agency (ESA), available online: https://earth.esa.int/eogateway (accessed on 18 July 2022).

**Acknowledgments:** The authors thank NSOAS for the HY-2B radar altimeter data, ASF for the Sentinel-1 SAR image data, ESA for the CryoSat-2 radar altimeter data.

**Conflicts of Interest:** The authors declare no conflict of interest.

## Appendix A

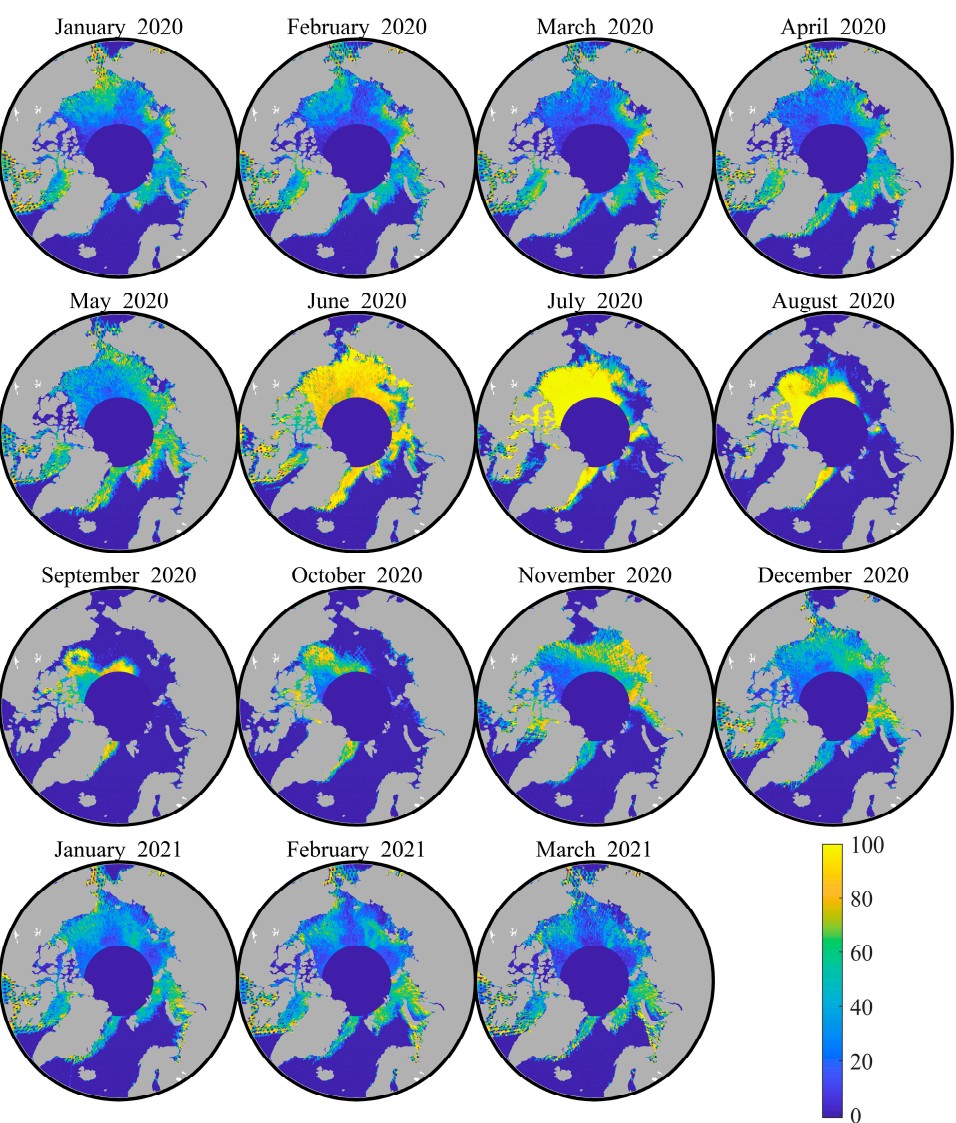

**Figure A1.** Lead fractions from January 2020 to March 2021 based on the HY-2B radar altimeter data and Bagging classifier.

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
