# Peer review of "Arctic Sea Ice Lead Detection from Chinese HY-2B Radar Altimeter Data"

_remotesensing, doi:10.3390/rs15020516_

Round 1
Reviewer 1 Report
The ms focuses on the analysis of ice extent in the Arctic region based on different data sources and applying different methodology. The results are presented in detail and compared against each other in terms of well defined statistical metrics. However, regarding metric selection and statistical analysis, several (minor) suggestions could be made:
1) To characterize the overlap between data/image sets, a common practice is to calculate two series of indices, the Intersection over Union (IoU) and the Dice-Sorensen coefficient (or simply Dice). The indices in Table 2 are indeed enough to calculate them, thus this should not be a problem.
2) Table 2 contains indices compared against certain ground truth data. However, it would be also interesting to look at cross-metrics quantifying the overlap between results obtained by either of the classifiers, such as IoU(i,j) and Dice(i,j) for all pairs of classifier methods (i,j) where i≠j.
3) Finally, statistical significance of those metrics have to be characterized, by indicating those pairs (i,j) that exhibit statistically significant differences at least in one of the metrics for a common confidence probability, e.g. p = 0.95. Given many values in Table 2 rather close to each other, this could eventually lead to the formation of clusters of classifier methods that do not actually result in significantly different estimates. Similarly, significance testing could be applied to the results against the ‘ground truth’. As a consequence, when no statistically significant differences could be observed within a certain cluster, there is no ground to claim that either of methods within this cluster appears either inferior or superior to the others, and vice versa.
The results of the statistical significance testing should be briefly commented and summarized in the Discussion and Conclusion, respectively.
Reviewer 2 Report
This manuscripts studies ten different surface type classification methods including supervised learning, unsupervised learning and threshold methods, being applied to the HY-2B radar altimeter data. Using the Sentinel-1 Synthetic Aperture Radar (SAR) images to verify, the supervised Bagging ensemble learning classifier has the best performance, and the ice lead has a reasonable spatial distribution and 21 is consistent with CryoSat-2 L2I data products.In general, the results of this manuscript are reliable, the overall framework is reasonable. However, there are still some minor content and textual issues that require further revision.
Line 116 The reason why only Ku-band radar altimeter data is used and not C-band should be explained here.
Figure2 The figure should indicate the spatial location of the SAR image matching the survey line, as depicted in Figure 3
Line 182 This should be Figure 4. You should check the full text for the figure number.
Section 2.2.1 Waveform characteristics are the key to the identification of leads, and the waveform characteristics of different types of surface returns should be described in more detail here, as well as a preliminary description based on the conclusions obtained in previous studies.
In addition, what is the basis for selecting these ten waveform parameters here?
Line 233 How the proportion of test and training datasets was selected and on what basis?
Line 226 How should this phrase be interpreted? Before training is to determine the number of leads?
‘this paper randomly selected the same amounts of samples as lead from sea ice and open water, respectively’.
Figure 5 Why only the results of K-means, other classification results should also be put together for comparison.
Line 348 Why 10km was chosen as the range threshold, as mentioned earlier only Ku-band data was used, what is the connection between this and C-band?
Section 3.3 It is suggested to switch the order of the subsections and arrange them according to supervised classification, unsupervised classification and threshold method, in line with the previous section
Section 3.4 CryoSat-2 uses SAR and SARIn modes for observations in the sea ice region, while HY-2B is a conventional radar, and how this affects the comparison results.
Figure 13/14/16/17 These figures need to be improved, try not to have text overlap, such as latitude markers.
In general, this manuscript has carried out a lot of work to compare and analyze various methods of inter-ice waterway identification, and it is recommended to revise and publish it, with particular attention to the review of text and figure names
Round 2
Reviewer 1 Report
The revised ms resolves previously raised issues and could be recommended for publication in its present form.
Reviewer 2 Report
The manuscript has been carefully revised according to the comments of reviewers, and the quality of the paper has been improved. I have no other comments and suggestions for the publication of this paper.